# Lightweight Detection of Silent Data Corruption in Distributed Deep Learning

## Abstract

Reliable detection of silent data corruption (SDC), such as bit-flip errors, is critical in large-scale neural network training, as undetected hardware faults can silently propagate and severely degrade model performance. We introduce a lightweight detection method integrated directly before collective communication steps, enabling localization of faulty devices with minimal runtime overhead. Our approach combines statistical modeling of gradient norms with divergence-based criteria to improve robustness. Experiments on large-scale training workloads, including LLaMA2 - 7B, show that our detector successfully identifies the vast majority of high-order bit-flip faults in bfloat16 while incurring only a very small computational overhead, offering a strong balance between detection accuracy and efficiency.

## 1 Introduction

As the size and complexity of machine learning models continue to grow, the probability of hardware-related failures and silent data corruption (SDC) Constantinescu et al. (2008) events increase substantially. Even single-bit errors, especially in high-order exponent or sign bits, can silently propagate through distributed training pipelines, leading to compromised model weights and unstable convergence Ghemawat et al. (2003), He et al. (2023). While some low-magnitude flips (e.g., mantissa $1 \to 0$) may be masked by optimizers, catastrophic flips often remain undetected. Traditional fallback strategies, such as complete checkpoint rollback or complete system restart, often prove costly and disruptive, especially for long-running, resource-intensive workloads involving hundreds or thousands of compute nodes Gemini Team (2025).

We propose a lightweight detector for SDC that is integrated into the communication layer of distributed training. By inserting checks before collective operations (e.g., all-reduce, all-gather), corrupted tensors can be identified before they propagate, enabling accurate device localization and real-time isolation. Our method combines multimodal detection of gradient norms with distance-based divergence metrics, achieving 99% detection accuracy with zero false positives at less than 0.5% runtime overhead in large-scale LLaMA2 - 7B training Touvron et al. (2023). We have implemented our solution in torch_npu adapter for Huawei Ascend devices, our approach requires no changes to high-level training code, offering a practical and scalable solution for reliable training.

## 2 Background

SDC denotes an undetected error in the output of a computing device, often triggered by hardware defects and highly dependent on instruction sequences Constantinescu et al. (2008). While early discussions focused on CPUs Ghemawat et al. (2003), modern heterogeneous ML stacks extend this risk to NPUs, and other accelerators. In practice, an affected device may silently miscompute even basic arithmetic without raising runtime exceptions, absent systematic software checks, such faults remain invisible to the training pipeline. This work adopts that a broader, device-agnostic view of SDC and focuses on its implications for large-scale, distributed training where undetected errors can propagate across nodes via collective communication.

## 2.1 BIT-FLIP

Machine learning models are typically trained with IEEE 754–compliant floating-point data types. In this format, even a single bit inversion can cause drastically different numerical effects depending on its position. Flipping low-order mantissa bits usually produces minor perturbations that optimizers and normalization layers can absorb, whereas altering high-order exponent or sign bits leads to severe deviations in magnitude, often resulting in NaN/INF values or numbers many orders of magnitude away from the original He et al. (2023). For example, flipping the leading exponent bit of a small float32 value can inflate it from $\approx 10^{-11}$ to $\approx 10^{28}$, while flipping a low mantissa bit changes the value by less than 1%. Figure 1 illustrates how inverting different bit positions systematically alters numerical values, with exponent flips producing particularly extreme distortions.

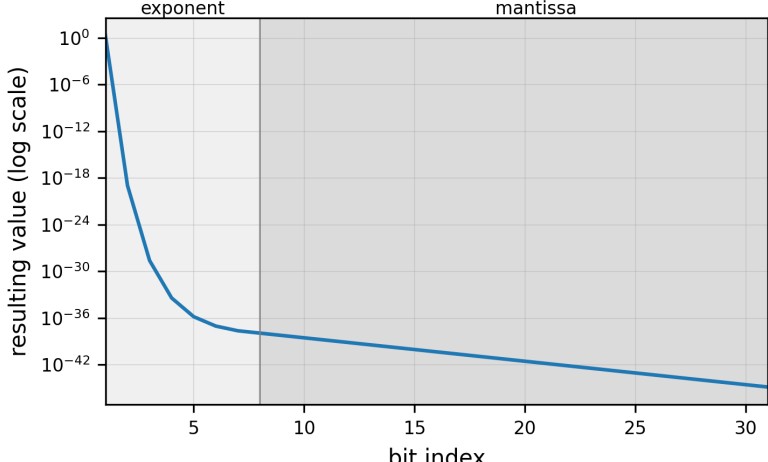

Figure 1: Impact of individual bit-flips in the IEEE 754 float32 representation of zero. The resulting values are plotted on a logarithmic scale. Shaded regions indicate mantissa and exponent fields.

The effect of a bit inversion can be expressed formally as

$$bff(x, i, n) = x \oplus 2^{n-i-1}, \tag{1}$$

where $x$ is the original value represented with $n$ bits, $i$ is the flipped bit position (indexed from left to right and started from 0), and $\oplus$ denotes bitwise XOR applied to the binary representation of $x$.

For bitflips in the exponent of a float number:

$$bff(x, i, n) = x \cdot 2^{2^{n-i-1}} \tag{2}$$

where $n$ is a number of bits in the exponent, $i$ is the position of the bit flip in the exponent.

To study aggregate effects, we apply (1) to float32 samples drawn from a standard normal distribution $x \sim \mathcal{N}(0, 1)$. Figure 2 shows the baseline distribution figure 2a alongside the corrupted one figure 2b, where 20% of the values have undergone a high-order exponent flip. The Gaussian bell shape is destroyed and the corrupted distribution exhibits pronounced heavy tails. An anomalous concentration of values near zero. These examples demonstrate a key property of SDC: high-order flips induce dramatic shifts that systematically distort statistical distributions.

## 2.2 DISTRIBUTED MODEL TRAINING

Distributed training refers to the process of training machine learning models across multiple computational units, often within one machine or across multiple machines in a cluster. The main objective is to accelerate training by leveraging the combined computational power and memory resources of multiple devices. There are many frameworks for distributed learning, such as Nagrecha (2023); Shoeybi et al. (2019); Narayanan et al. (2021); Korthikanti et al. (2022); Liu et al. (2023); Sergeev & Balso (2018), etc. However, with an increasing number of devices involved in training, the likelihood

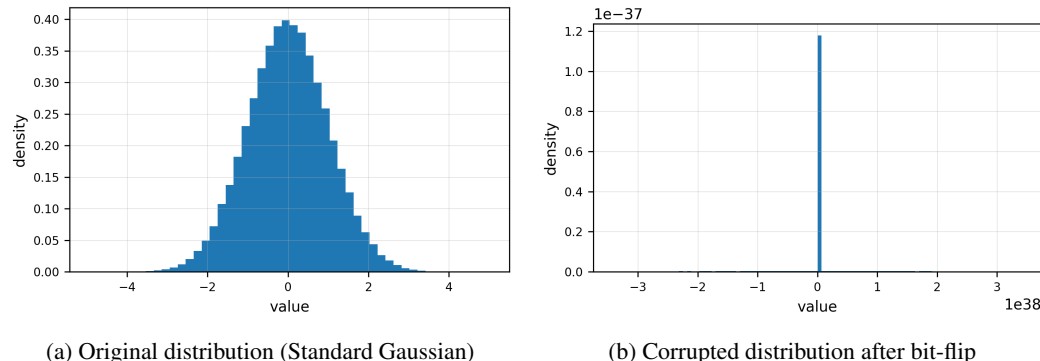

(a) Original distribution (Standard Gaussian)   (b) Corrupted distribution after bit-flip

Figure 2: Distributional impact of a high-order bit-flip: (a) baseline $\mathcal{N}(0,1)$ and (b) corrupted distribution after flipping the exponent bit in 20% of the samples.

of encountering hardware failures also grows, due to the added complexity and higher probability of component malfunctions across multiple units.

Even if a device appears healthy, it can still receive corrupted parameters during collective operations that merge outputs or gradients from all nodes in the cluster Ma et al. (2025). Because these updates are broadcast to and from every participant, a single node with silent data corruption can effectively contaminate otherwise healthy devices. This process unfolds through collective communication primitives (like all-reduce, all-gather, or reduce-scatter), which unify the outputs from all nodes and integrate them back into the shared model state. As a result, once an error is introduced in one device's computations, it can spread to the entire training cluster, even reaching nodes that were initially free of faults.

## 2.3 SDC IMPACT IN THE LEARNING PROCESS

Silent data corruption (SDC) typically manifests as bit-flips, i.e., random changes to individual bits in memory or during data transfers. He et al. (2023) investigated hardware failures in accelerator-based deep learning systems and found that training is most severely impacted by high-magnitude errors. Such errors can lead to a variety of outcomes, including:

- gradual degradation of accuracy,
- an abrupt accuracy drop followed by slow degradation,
- an instantaneous and severe accuracy collapse,
- training that appears normal but yields significantly reduced final accuracy.

These effects are often caused by transient faults or circuit degradation in accelerator logic. Some errors can be mitigated by scaling mechanisms inherent in normalization layers and optimizers that exploit gradient history, which partially compensates for numerical deviations and prevents their accumulation.

Nevertheless, detecting hardware-induced errors remains essential, as many can cause serious disruptions in the training process. Particularly critical are unrecoverable faults, such as sudden jumps producing NaN/INF values or corrupted weight updates, which can render the model unusable. Even errors that may eventually be compensated for by normalization or optimization still waste computational resources: they degrade convergence, temporarily reduce accuracy, and increase training cost. Timely detection and correction of such faults are therefore crucial for improving both the reliability and efficiency of large-scale training.

Liang et al. (2025) further studied hardware-induced faults in the attention mechanism of large language models, including BERT Devlin et al. (2018), GPT-2 Tan et al. (2020), GPT-Neo Kashyap et al. (2023), and RoBERTa Liu et al. (2019). The attention module is particularly vulnerable due to its computational complexity. Bit-flips in floating-point computations can introduce INF, NaN,

or near-INF values, which then propagate through key operations, such as matrix multiplication (GEMM), softmax, and normalization. This makes attention-related failures especially dangerous, as they can compromise the stability of the entire training pipeline.

## 3 APPROACH

### 3.1 MULTIMODALITY EMERGENCE IN TRAINING UNDER SDC INFLUENCE

Multimodality in the distribution of optimizer parameters or gradient statistics is often associated with transient phenomena during training and may indicate instability, such as sudden gradient spikes Molybog et al. (2023). Although such occurrences are relatively rare, they may also arise naturally from architectural factors (e.g., skip connections, multi-head attention) or optimization dynamics. Nevertheless, the unexpected appearance of multimodality provides a strong signal for detecting potential pathologies in the learning process.

Our approach focuses on analyzing the norms of the gradients after a logarithmic transformation, i.e., studying $\log(\|\nabla L\|)$. If the raw gradient norms form a mixture of values with substantially different scales, such as those induced by anomalously large updates due to SDC, the distribution may still appear unimodal or only weakly bimodal. Applying the logarithmic transformation amplifies these scale differences, causing the modes to separate and thereby making multimodality more pronounced.

**Theorem 1 (Violation of Unimodality under High-Order Bit-Flip Corruption.)** *Let $X$ be a real-valued random variable with probability density function $f_X$ and $\Pr(X = 0) = 0$. Assume that $Y = \log_2(|X|)$ is unimodal. Consider the corrupted variable $G$ obtained by applying the bit-flip transformation equation 1 with probability $p \in (0, 1)$ in the exponent of a number with respect to IEEE 754:*

$$G = \begin{cases} \log_2(|X|) + \Delta, & \text{with probability } p, \\ \log_2(|X|), & \text{with probability } 1 - p, \end{cases}$$

*where $\Delta = 2^{n-i-1}$ denotes the flipped bit position in the IEEE 754 representation of $X$. Then there exists a critical position $\Delta_{\mathrm{crit}}(p)$ such that for all $\Delta \geq \Delta_{\mathrm{crit}}(p)$, the distribution of $G$ is multimodal.*

A detailed proof of this theorem is provided in appendix B.1. The result formalizes the intuition that SDC, modeled as high-order bit-flips in floating-point values, inevitably creates distinct clusters in the log-transformed domain, thus violating unimodality. This provides a theoretical foundation for using multimodal behavior as a statistical indicator of anomalies during training. In this work, we used $\Delta_{crit} \approx 14.36$, which corresponds to the third bit of the exponent, i.e., we focused on detecting errors that occurred in the first three bits of the exponent. The conclusion can be found in appendix B.2.

Multimodality in gradient statistics does not necessarily imply the presence of SDC. Such patterns may also arise naturally from model dynamics, optimization artifacts, or data heterogeneity. Still, when the modes are well separated, i.e., when the inter-cluster distance is large, the emergence of multimodality can serve as a useful indicator of potential corruption. In practice, combining this signal with distance-based metrics further improves the reliability of SDC detection in large-scale training pipelines.

### 3.2 DIAGNOSING MULTIMODAL BEHAVIOR IN TRAINING DYNAMICS

As established in the previous section, SDC can induce multimodality in the distributions of training-related statistics, such as gradient norms. Detecting such multimodal patterns in training dynamics therefore provides a valuable signal of potential hidden errors. Several statistical tests have been proposed for diagnosing multimodality, each offering different trade-offs in sensitivity, robustness, and computational cost. Classical approaches such as Hartigan's Dip Test Hartigan & Hartigan (1985) and Silverman's Test Silverman (1981) provide strong theoretical guarantees but are computationally expensive, often requiring bootstrapping. More recent methods, such as the UU-Test Chasani & Likas (2022), are robust to noise but less established in practice. On the other hand, the Folding Test Siffer et al. (2018) has been demonstrated to be effective when multimodality arises from symmetric deviations and has linear-time complexity, making it particularly well-suited for large-scale distributed training. A detailed comparison of these tests, including their advantages and

limitations, is provided in Appendix A. In the remainder of this work, we adopt the Folding Test as the basis for multimodality detection, as it provides the best balance between computational efficiency and detection reliability in our setting.

## 3.3 METRICS FOR DISTRIBUTION SHIFT INDUCED BY GRADIENT CORRUPTION

As established in theorem 1, high-order bit-flips caused by SDC produce additional modes in the distribution of gradient norms, with corrupted values displaced by several orders of magnitude from the clean ones. Such anomalies are meaningful only if the emerging modes are sufficiently separated, creating a measurable distributional shift rather than minor local fluctuations. To evaluate this effect, one requires a metric that captures both the presence of extra modes and the degree of their separation from the original distribution.

For this purpose, we adopt the Wasserstein distance Villani (2008) (Earth-Mover distance), which offers favorable theoretical and practical properties for detecting gradient corruption. In contrast to divergence-based measures such as Kullback - Leibler Kullback & Leibler (1951) or Jensen - Shannon Lin (1991), which may become undefined or unstable when distributions have disjoint support, the Wasserstein distance is a true metric. It measures the minimal transportation cost required to transform one distribution into another, and thus remains well defined when SDC shifts gradients into previously unoccupied regions.

Moreover, the Wasserstein distance is sensitive not only to the emergence of multiple modes but also to their displacement, capturing both local and global aspects of distributional divergence. This sensitivity is critical for distinguishing genuine corruption-induced shifts from benign variability in training dynamics.

For one-dimensional empirical distributions $P = \{x_1, \ldots, x_n\}$ and $Q = \{y_1, \ldots, y_n\}$, sorted in increasing order, the first-order Wasserstein distance is given by

$$W_1(P, Q) = \frac{1}{n} \sum_{i=1}^{n} |x_i - y_i|, \tag{3}$$

which can be computed in linear time after sorting. This efficiency makes the Wasserstein distance particularly suitable for online diagnostics of distributional shifts during large-scale training.

## 3.4 CONSTRUCTION OF THE SDC DETECTION PIPELINE

To detect SDC during training, we propose a lightweight statistical pipeline based on the analysis of gradient norms. Algorithm 1 describes the entire pipeline of the SDC detection. The detection process consists of three stages:

1. **Logarithmic transformation.** For a batch of gradients $G = \{g_1, g_2, \ldots, g_m\}$, with $g_i \in \mathbb{R}^n$, we compute

$$V = \{\log \|g_1\|, \log \|g_2\|, \ldots, \log \|g_m\|\} \in \mathbb{R}^m.$$

This step normalizes scale differences and enhances separation between potential modes.

2. **Unimodality assessment.** The distribution of $V$ is evaluated with the Folding Test. If the test accepts unimodality, the batch is considered clean. Complexity: $O(m)$.

3. **Intermodal separation.** If multimodality is detected, the Folding Test provides a pivot point $s^* \in \mathbb{R}$ between the modes. We partition

$$V_{\text{left}} = \{v_i \in V : v_i \leq s^*\}, \quad V_{\text{right}} = \{v_i \in V : v_i > s^*\},$$

and compute the Wasserstein distance

$$W_1(V_{\text{left}}, V_{\text{right}}) = \int |F_{V_{\text{left}}}(x) - F_{V_{\text{right}}}(x)| dx,$$

where $F_{V_{\text{left}}}, F_{V_{\text{right}}}$ are cumulative density functions for $V_{\text{left}}$ and $V_{\text{left}}$ correspondingly. If $W_1 > \tau$, with $\tau$ an empirically chosen threshold, the shift is deemed significant and SDC is flagged.

Each stage is computationally inexpensive ($O(m \log m)$ overall, dominated by sorting for Wasserstein distance), enabling integration into large-scale distributed training with negligible overhead.

**Algorithm 1** SDC detection pipeline

**Require:** Gradients $G = \{g_1, \ldots, g_m\}$, threshold $\tau$
**Ensure:** Detection flag
 1: $V \leftarrow \{\log \|g_1\|, \ldots, \log \|g_m\|\}$
 2: $(s^*, \text{is\_unimodal}) \leftarrow \texttt{FoldingTest}(V)$
 3: **if** is_unimodal **then**
 4:     **return** No SDC detected
 5: **end if**
 6: Partition $V$ into $V_{\text{left}} = \{v_i < s^*\}$, $V_{\text{right}} = \{v_i > s^*\}$
 7: $d \leftarrow W_1(V_{\text{left}}, V_{\text{right}})$
 8: **if** $d > \tau$ **then**
 9:     **return** SDC detected
10: **else**
11:     **return** No SDC detected
12: **end if**

## 4    IMPLEMENTATION

The proposed SDC detection pipeline was implemented on a cluster of 8 Huawei Ascend 910B devices. Each device combines specialized processing units, including AI Cores for dense linear algebra, AI CPUs for control logic, and dedicated memory subsystems. The overall processor organization is illustrated in Figure 3.

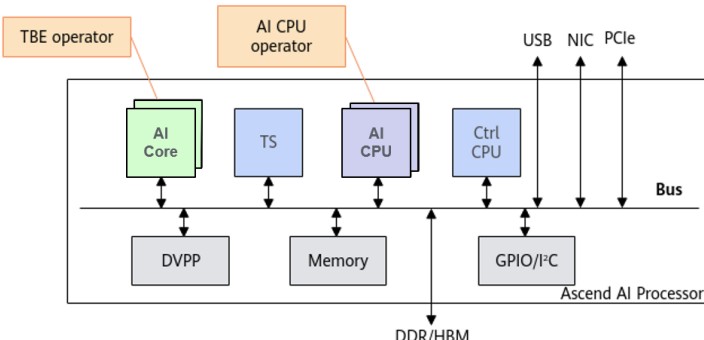

Figure 3: placement of the custom CANN operator for SDC detection within the Ascend AI processor.

In our implementation, computational tasks were distributed between the AI Core and AI CPU according to their architectural strengths:

- **AI Core.** The computation of logarithmic gradient norms was executed on the AI Core using custom Tensor Boost Engine (TBE) kernels, leveraging its high-throughput vector and matrix processing capabilities.

- **AI CPU.** Multimodality detection (Folding Test) and intermodal Wasserstein distance estimation were performed on the AI CPU. These operations involve control logic, for which the AI CPU is better suited.

The AI CPU operations were scheduled asynchronously and overlapped with AI Core execution, enabling pipelined processing and minimizing additional latency. This division of labor ensures that both dense numeric kernels and control-heavy statistical tests are executed on hardware optimized for their respective workloads.

To minimize integration effort, the SDC checker was embedded at the communication layer of the `torch_npu` adapter. Specifically, checks were inserted immediately before inter-device communication, allowing detection of corrupted tensors prior to their propagation across nodes. This

design avoids any modification to user-level training code while ensuring compatibility with existing distributed training pipelines.

## 5 EXPERIMENTS

### 5.1 FAULT INJECTOR

To evaluate the robustness of our SDC detection method under controlled conditions, we implemented a custom fault injection simulator. The simulator introduces single- and multi-bit flips into floating-point values across different model states. This design enables a systematic assessment of the sensitivity of the detection pipeline to diverse corruption patterns. The pseudocode of the bit-flip injection procedure is shown in Algorithm 2.

Faults were injected by modifying the byte-level representation of numerical values, flipping the specified bit positions, and reconstructing the corrupted values. In our experiments, bit positions were selected uniformly at random from the most significant portion of the exponent field. This choice reflects the fact that high-order bit flips have the strongest impact on numerical magnitude, often shifting values by many orders. Such errors are thus most likely to induce distributional shifts detectable by our method, whereas low-order flips typically behave as minor noise and do not materially affect training dynamics.

To ensure reproducibility, injections were performed on complete model dumps obtained from training checkpoints. This allowed us to target precise numerical values within the model state and emulate realistic corruption scenarios during training. The resulting setup provides a controlled yet representative environment for evaluating both the accuracy and computational overhead of the proposed SDC detection pipeline.

---

**Algorithm 2** Bit-flip fault injection algorithm

---

**Require:** Numerical value $v$, bit positions $\mathcal{P}$
**Ensure:** Corrupted numerical value $v'$
1: Compute byte representation $B \leftarrow \text{bytes}(v)$
2: **for** each position $p \in \mathcal{P}$ **do**
3:      Identify byte index $q$ and bit index $r$: $q, r \leftarrow \text{divmod}(p, 8)$
4:      Flip bit $r$ of byte $q$: $B[q] \leftarrow B[q] \oplus (1 \ll r)$
5: **end for**
6: Reconstruct corrupted value $v'$ from modified bytes $B$
7: **return** $v'$

---

### 5.2 RESULTS

We evaluate the detector during LLaMA2 - 7B training under controlled single-bit exponent flips. We inject 300 faults in total, uniformly across three flip types (100 each), where type-1/2/3 means flipping the 1st/2nd/3rd bit of the IEEE 754 exponent, respectively. Table 1 reports the per-type breakdown at a fixed operating point ($\tau = 3$ for the Wasserstein test): overall, 297/300 faults are detected (99.0% TPR) with zero false positives; type-1/2 reach 100% TPR, and all three misses come from type-3. The false negatives arise because:

- the perturbation is attenuated before the pre-collective check by optimizer dynamics and normalization/residual mixing,

- the residual shift at test time remains within natural stochastic variability no clear multi-modality or a Wasserstein distance below $\tau$.

The resulting time overhead is $\sim 0.46\%$ (86,890.60 ms per iteration for baseline vs. 87,294.26 ms with the detector). Figure 4 illustrates the distributional signal used by the detector: corrupted batches exhibit separated clusters, while missed cases stay within the baseline envelope.

Table 1: Per-type SDC detection on LLaMA2 - 7B ($n$=100 per type; total 300) at fixed threshold $\tau$=3.

| Bit-flip type | Injected | Detected (TP) | False Pos. | False Neg. | TPR |
|---|---|---|---|---|---|
| Type-1 | 100 | 100 | 0 | 0 | 100.0% |
| Type-2 | 100 | 100 | 0 | 0 | 100.0% |
| Type-3 | 100 | 97 | 0 | 3 | 97.0% |
| **Total** | **300** | **297** | **0** | **3** | **99.0%** |

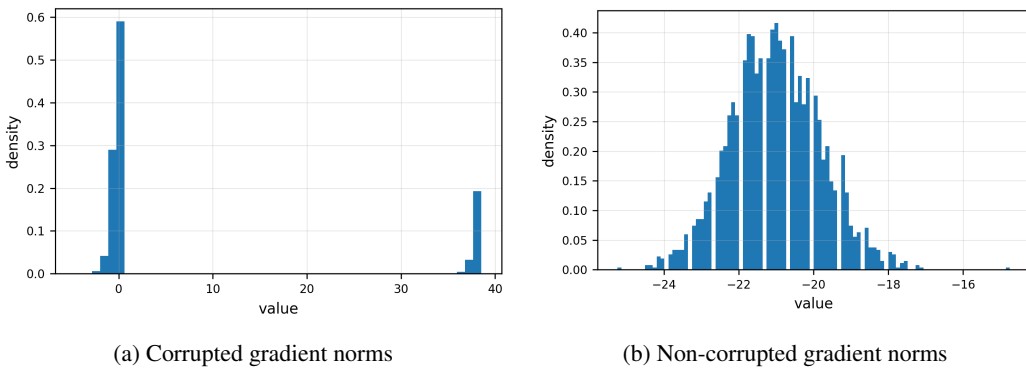

(a) Corrupted gradient norms

(b) Non-corrupted gradient norms

Figure 4: Gradient norm distributions with and without injected bit-flips.

## 6 RELATED WORKS

Recent research has highlighted the critical challenges posed by hardware failures and silent data corruption (SDC) in large-scale deep neural network (DNN) and large language model (LLM) training. Ma et al. Ma et al. (2025) provided one of the first systematic investigations, showing that bit-flips at critical gradient or activation stages can lead to subtle but irreversible parameter drift. Their lock-step synchronization mechanism partially mitigates these effects, but at the cost of tight coupling between training processes.

He et al. He et al. (2023) conducted large-scale fault injection studies on accelerator hardware, demonstrating that even small defects in memory or logic pathways can propagate through linear algebra operations and compromise model accuracy. They further showed that naive checkpointing becomes prohibitively expensive under frequent failures, motivating the search for lightweight alternatives.

At the system level, Wu et al. Wu et al. (2023) proposed the TRANSOM framework, which combines adaptive checkpointing, anomaly detection, and on-the-fly rescheduling to manage node-level faults. While effective in large clusters, this approach requires extensive monitoring infrastructure and incurs non-negligible coordination overhead.

Liang et al. Liang et al. (2025) focused on the transformer attention mechanism and developed *ATTNChecker*, a submodule-level method that detects INF/NaN values and applies algorithm-based fault tolerance (ABFT) directly within matrix multiplications. Their approach achieves high accuracy in intercepting catastrophic errors, but remains specialized to a single model component and does not address broader distributional corruptions.

Taken together, prior work highlights two dominant directions: submodule-specific resilience mechanisms (e.g., synchronization, ABFT) and system-level frameworks (e.g., dynamic checkpointing, scheduling). Both classes of methods can be effective, but they either impose substantial runtime overhead or require intrusive architectural modifications. In contrast, our method operates at the communication boundary, introducing a lightweight statistical detector that requires no model-specific

changes and incurs only ∼2% overhead. This positions our approach as a complementary strategy that bridges the gap between fine-grained local protection and global system-level recovery.

## 7 CONCLUSION

We presented a lightweight statistical pipeline for detecting silent data corruption (SDC) during large-scale neural network training. By combining logarithmic transformation of gradient norms, unimodality assessment via the Folding Test, and intermodal separation with the Wasserstein distance, our method enables efficient online detection of corruption events. A key advantage of this approach is its integration at the communication layer, which allows faulty devices to be localized without modifying model code or disrupting the training pipeline. Experimental evaluation on LLaMA2 - 7B demonstrated a 99% detection rate with no false positives and only ∼2% runtime overhead.

At the same time, the current study has several limitations. First, the decision threshold for the Wasserstein distance could not be derived analytically and was instead determined empirically (set to $\tau = 3$ for LLaMA2-7B). Second, small-magnitude bit-flips, which do not significantly affect model convergence, remain outside the scope of our detection. Finally, evaluation was limited to a single model architecture; broader validation is required to assess generality.

Future work will explore adaptive thresholding strategies, integration with complementary resilience mechanisms (e.g., algorithm-based fault tolerance in attention layers), and experiments across diverse architectures and training scales. Together, these directions aim to extend the robustness and applicability of the proposed approach, providing a foundation for resilient large-scale training under increasingly heterogeneous and failure-prone hardware environments.

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

## A   COMPARISON OF MULTIMODALITY TESTS

Table 2 summarizes several commonly used statistical tests for detecting multimodality. This extended comparison complements the discussion in Section 3.2, where the Folding Test was chosen as the most suitable option for our setting.

Table 2: comparison of statistical tests for multimodality detection

| test name | description | advantages | disadvantages | complexity |
|---|---|---|---|---|
| Hartigan's Dip Test Hartigan & Hartigan (1985) | Measures maximum deviation of the empirical CDF from the closest unimodal distribution. | General-purpose; parameter-free; widely used. | Sensitive to noise; bootstrapping needed for p-values. | $\mathcal{O}(n \log n)$ |
| Silverman's Test Silverman (1981) | KDE + boot-strapping to test the number of modes. | Highly sensitive; can test for a specific number of modes. | Bandwidth/kernel dependent; computationally expensive. | $\mathcal{O}(n^2 + Bn)$ |
| Folding Test Siffer et al. (2018) | Compares variance of the "folded" distribution to the original. | Fast; simple; effective for symmetric deviations. | Less sensitive to asymmetry; no p-value. | $\mathcal{O}(n)$ |
| UU-Test Chasani & Likas (2022) | Approximates the empirical CDF with uniform segments. | No bootstrapping; parameter-free; robust to noise. | Requires sorting; relatively new/less validated. | $\mathcal{O}(n \log n)$ |

## B   PROOFS

### B.1   PROOF OF THE THEOREM 1

**Representation of the density** $G$**.** By the law of total probability, the density $f_G$ is a mixture of two components:
$$f_G(z) = (1-p)f_Y(z) + pf_Y(z - \Delta), \quad z \in \mathbb{R}.$$
Both functions $f_Y(z)$ and $f_Y(z - \Delta)$ are continuous and differentiable; therefore, $f_G$ is also continuous and differentiable everywhere.

**Sign of the derivative around the mode of the original density.** Let $m$ be the (unique) mode of $f_Y$. By unimodality we have
$$f_Y'(z) > 0 \quad \text{for } z < m, \quad f_Y'(m) = 0, \quad f_Y'(z) < 0 \quad \text{for } z > m.$$

Consider the derivative of the mixture:
$$f_G'(z) = (1-p)f_Y'(z) + pf_Y'(z - \Delta).$$

Now compute $f_G'$ at two key points, $z = m$ and $z = m + \Delta$.

At $z = m$:
$$f_G'(m) = (1-p)f_Y'(m) + pf_Y'(m - \Delta) = 0 + pf_Y'(m - \Delta).$$
Since $m - \Delta < m$, by unimodality we have $f_Y'(m - \Delta) > 0$. Thus,
$$f_G'(m) = pf_Y'(m - \Delta) > 0.$$

At $z = m + \Delta$:
$$f_G'(m + \Delta) = (1-p)f_Y'(m + \Delta) + pf_Y'(m) = (1-p)f_Y'(m + \Delta) + 0.$$
Since $m + \Delta > m$, we have $f_Y'(m + \Delta) < 0$, hence
$$f_G'(m + \Delta) < 0.$$

Therefore, $f'_G(m) > 0$ and $f'_G(m + \Delta) < 0$. Therefore, $f'_G(m) > 0$ and $f'_G(m + \Delta) < 0$.

Since $f'_G$ is continuous, the change of sign between $m$ and $m + \Delta$ guarantees the existence of a point

$$c \in (m, m + \Delta) \quad \text{such that } f'_G(c) = 0.$$

Thus, inside the interval $(m, m + \Delta)$ there exists a stationary point of $f_G$.

**Second derivative analysis and threshold** $\Delta_{\text{crit}}(p)$. Let $z_0$ be a stationary point of $f_G$, i.e. $f'_G(z_0) = 0$. The Taylor expansion of $f_G$ about $z_0$ reads

$$f_G(z) = f_G(z_0) + \tfrac{1}{2}f''_G(z_0)(z - z_0)^2 + \tfrac{1}{3!}f_G^{(3)}(z_0)(z - z_0)^3 + \tfrac{1}{4!}f_G^{(4)}(z_0)(z - z_0)^4 + \cdots.$$

The leading second-order term determines the local quadratic behaviour:

- If $f''_G(z_0) < 0$ and higher-order terms are negligible, then $z_0$ is a (strict) local maximum and the graph near $z_0$ resembles an inverted parabola.

- If $f''_G(z_0) > 0$, then $z_0$ is a local minimum.

- If $f''_G(z_0) = 0$, the quadratic term vanishes and the local shape is governed by higher-order terms. For example:

    - If $f_G^{(3)}(z_0) \neq 0$, the cubic term dominates and $z_0$ is an inflection point with asymmetric cubic behaviour.
    - If $f_G^{(3)}(z_0) = 0$ but $f_G^{(4)}(z_0) < 0$, then the quartic term dominates and a flattened local maximum/plateau may appear (typical for symmetric profile around $z_0$).

**Definition of the critical shift** $\Delta_{\text{crit}}$. A natural criterion for the loss of unimodality is the emergence of a stationary point $z^* \in (m, m + \Delta)$ at which the second derivative changes sign from non-positive to non-negative. Thus we define $\Delta_{\text{crit}}(p)$ as the smallest shift $\Delta > 0$ for which there exists a point $z^* \in (m, m + \Delta)$ such that

$$f'_G(z^*) = 0 \quad \text{and} \quad f''_G(z^*) = 0.$$

The first condition identifies $z^*$ as a stationary point; the second marks the transition in the character of the stationary point (a bifurcation boundary between "no interior minimum" and "interior minimum present").

When $\Delta < \Delta_{\text{crit}}(p)$ any stationary point inside $(m, m + \Delta)$ is either absent or corresponds to a point where $f''_G \leq 0$, so the mixture remains effectively unimodal (the peaks overlap). For $\Delta > \Delta_{\text{crit}}(p)$ the stationary point splits into a strict local minimum flanked by two distinct local maxima (one near $m$, the other near $m + \Delta$), i.e. the mixture becomes multimodal. Thus, the theorem is proved.

### B.2  CONCLUSION $\Delta_{crit}$ FOR THE CASE OF NORMAL DISTRIBUTION

We consider that $|X|$ has like lognormal distribution. And $Y = \log_2(|X|)$ is normal distribution then:

$$\phi_\sigma(y) = \frac{1}{\sigma\sqrt{2\pi}} e^{-\frac{-y^2}{2\sigma^2}}.$$

Then:

$$\phi'(y) = -\frac{y}{\sigma^2}\phi(y), \quad \phi''(y) = \frac{y^2 - \sigma^2}{\sigma^4}\phi(y).$$

The condition for the critical point $y^*$ is determined by the system:

$$\begin{cases} (1 - p)\phi'(y^*) + p\phi'(y^* - \Delta) = 0, \\ (1 - p)\phi''(y^*) + p\phi''(y^* - \Delta) = 0. \end{cases}$$

From the first we obtain:

$$\phi(y^* - \Delta) = \frac{(1 - p)y}{p(\Delta - y^*)}\phi(y^*).$$

From the second we obtain:

$$(1 - p)(y^{*2} - \sigma^2)\phi(y^*) + p((y^* - \Delta)^2 - \sigma^2)\phi(y^* - \Delta) = 0.$$

After expanding the brackets and simplifying some terms, we obtain:

$$(1-p)\left(-\frac{y^*}{\sigma^2}\phi(y^*)\right) + p\left(-\frac{y^*-\Delta}{\sigma^2}\phi(y^*-\Delta)\right) = 0$$

$$(1-p)y^*\phi(y^*) - p(\Delta - y^*)\phi(y^*-\Delta) = 0$$

$$\frac{(1-p)y^*}{p(\Delta - y^*)} = \frac{\phi(y^*-\Delta)}{\phi(y^*)}.$$

Calculate the ratio $\frac{\phi(y^*-\Delta)}{\phi(y^*)}$. Since $\phi(y) \propto e^{-\frac{y^2}{2\sigma^2}}$, we have:

$$\frac{\phi(y^*-\Delta)}{\phi(y^*)} = \exp\left(-\frac{(y^*-\Delta)^2}{2\sigma^2} + \frac{y^{*2}}{2\sigma^2}\right) = \exp\left(\frac{y^{*2}-(y^*-\Delta)^2}{2\sigma^2}\right) =$$

$$= \exp\left(\frac{2z^*\Delta - \Delta^2}{2\sigma^2}\right).$$

Then:

$$\ln\left(\frac{(1-p)y^*}{p(\Delta - y^*)}\right) = \frac{y^*\Delta - \frac{1}{2}\Delta^2}{\sigma^2}.$$

As a result, we obtain that $\Delta_{crit}(p,\sigma)$ is defined as the minimum $\Delta \geq 2\sigma$ such that:

$$\ln\left(\frac{(1-p)y^*}{p(\Delta - y^*)}\right) = \frac{y^*\Delta - \frac{1}{2}\Delta^2}{\sigma^2},$$

when $y^* = \frac{\Delta \pm \sqrt{\Delta^2 - 4\sigma^2}}{2}$ and $p$ is given. Figure 5 shows the values of $\Delta_{crit}$ depending on $p$.

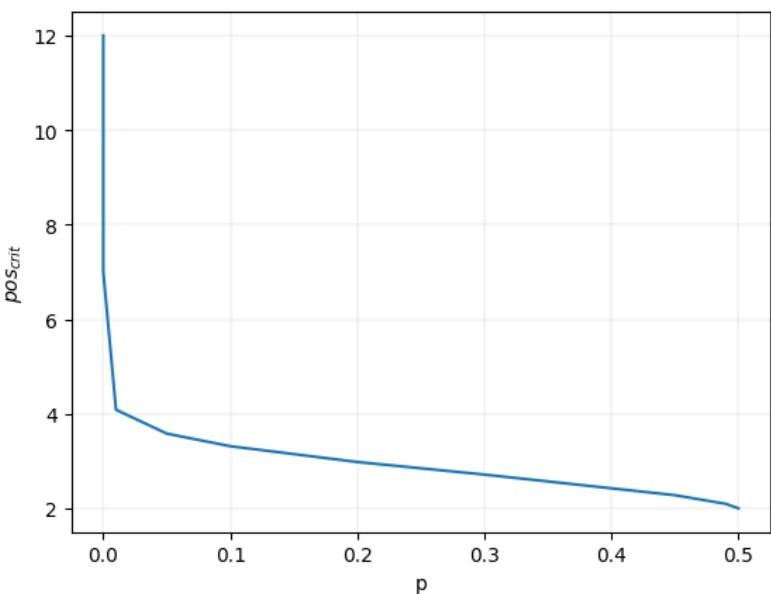

Figure 5: Dependence of $\Delta_{crit}$ on $p$ when $\sigma=1$
.

In this work, we use $\Delta_{crit} = 14.36$ because the probability of a bit-flip occurring is not that high in reality, and we have derived a theoretical upper limit, but we also check communications, and before the error reaches the communication, it will have time to smooth out. From equation 2, this corresponds to the third bit of the exponent.

We examined various tensors at $p = 1e^{-10}$ for which we calculated $\Delta_{crit}$ and **position** $= n - 1 - \log_2 \Delta_{crit}$; the results are shown in the table 3.

Table 3: $\Delta_{crit}$ for various tensors of the LLaMA2 - 7B model for $p = 1e^{-10}$

| $\sigma$ | $\Delta_{crit}$ | position | shapes |
|---|---|---|---|
| 2.42 | 18.14 | 2.82 | $1024 \times 4 \times 2048$ |
| 1.99 | 14.92 | 3.10 | $2560 \times 2560$ |
| 1.72 | 12.87 | 3.31 | $4096 \times 2 \times 480$ |
| 1.81 | 13.57 | 3.24 | $4096 \times 2 \times 480$ |
| 1.79 | 13.40 | 3.26 | $4096 \times 2 \times 480$ |
| 1.77 | 13.28 | 3.27 | $4096 \times 2 \times 480$ |

# C  USE OF LARGE LANGUAGE MODELS (LLMS)

**Writing assistance.** We used a large language model (OpenAI ChatGPT) strictly as a copy-editing aid to improve grammar, clarity, concision, and consistency of terminology. No mathematical statements, algorithms, experimental designs, or conclusions were authored by the LLM; all technical content and structure originated from the authors.

**Retrieval and discovery.** We used the LLM to help discover and organize related work by generating candidate paper lists and topical groupings from our queries. All suggested references were independently verified by the authors for accuracy and relevance, and any unverifiable citations were discarded. Final literature selection and summaries were written and vetted by the authors.

**Accountability and authorship.** The authors take full responsibility for the paper's contents. The LLM is not an author and did not contribute intellectual novelty or research ideation.

**Data handling.** We did not share proprietary data, unpublished results, or personal information with the LLM beyond brief, high-level paraphrases necessary for editing. All experimental results, analyses, figures, and tables were produced and checked by the authors.

**Plagiarism and factual checks.** LLM-edited text was reviewed line-by-line by the authors. Technical claims, equations, and citations were verified against primary sources and our own experiments.

