# OpenReview forum: "Lightweight Detection of Silent Data Corruption in Distributed Deep Learning"
_ICLR.cc/2026/Conference — Submitted to ICLR 2026_

### Official Review · Reviewer_CurN · 2025-10-31

**Soundness:** 1
**Presentation:** 3
**Contribution:** 1
**Rating:** 0
**Confidence:** 5

**Summary:**

This work presents a statistical method for lightweight detection of SDCs (Silent Data Corruption) in distributed neural network training. This is done by checking for multimodality in the distribution of NN gradients - if the gradient distribution is unimodal, the test passes and SDCs are assumed to not exist. In the event of the test showing multimodality, the Wasserstein distance at the pivot point of the set of gradients (the distance between these halves) is used to check for significant distribution distance. This is then used to flag the presence of an SDC. This approach is asymptotically computationally cheap, scaling as O(mlogm).

Having worked on SDCs for quite a while in the hardware space, I have several questions about this approach:
1) There has been extensive work on low-cost algorithm-based fault tolerance in the hardware space. Algorithmic checksums have been shown theoretically to achieve O(1) correction of single bit errors in linear matrix multiplications [1], and that same paper illustrates prior art dating back to the  1980s implementing these checksums in high-performance computing. More recent work has achieved very high resilience using simpler statistical tests on neuron statistics in reinforcement learning training [2]. How is this work distinct and more importantly scalable, given the greater sophistication of Wasserstein distance computations and the folding test? Citations of prior art and contrasts with prior art are needed.
2) On that note, the current work pays attention to all-reduce operations and communications between work units in training, hence focusing on gradient values. Ozen et al [3] have achieved extremely low cost resilience in the forward pass using linear algorithmic checksums, and by setting neuron values to zero in inference [4]. Given that this may also work for forward pass computations in training, one can regard the forward pass as secure? If so, are the bitflips injected here into backward pass computations? I do not see clear locations of injection in the training flow, and contrast with prior work.
3) How does this approach contrast with other hardware-focused fault tolerance mechanisms such as resilience-aware scheduling, redundancy, algorithmic noise tolerance? The related work section ignores a great deal of prior art.
4) Lastly, I do not see a clear noise model, or an accurate one. I quote, "We inject 300 faults in total, uniformly across three flip types (100 each), where type-1/2/3 means flipping the 1st/2nd/3rd bit of the IEEE 754 exponent, respectively." Ozen et al have shown in [5] that bit error rates are as low as 1e-4 or 1e-5 in real operation, and this is distributed across the entire IEEE754 word, uniformly - not just the most significant bits. As such, how does this noise model, simply flipping the most significant exponent bits of the word, in any way test the sensitivity of this framework? Surely randomized injection corresponding to a probability of bit error would be more realistic?

[1]  Franklin T. Luk and Haesun Park "An Analysis Of Algorithm-Based Fault Tolerance Techniques", Proc. SPIE 0696, Advanced Algorithms and Architectures for Signal Processing I, (4 April 1986); https://doi.org/10.1117/12.936896

[2] C. Amarnath, M. Mejri, J. Isenberg and A. Chatterjee, "Error Resilient Online Reinforcement Learning Using Adaptive Statistical Checks," in IEEE Transactions on Computer-Aided Design of Integrated Circuits and Systems, vol. 44, no. 8, pp. 3112-3125, Aug. 2025, doi: 10.1109/TCAD.2025.3529820

[3] Ozen, E., Orailoglu, A. Low-Cost Error Detection in Deep Neural Network Accelerators with Linear Algorithmic Checksums. J Electron Test 36, 703–718 (2020). https://doi.org/10.1007/s10836-020-05920-2

[4] Elbruz Ozen and Alex Orailoglu. 2020. Just say zero: containing critical bit-error propagation in deep neural networks with anomalous feature suppression. In Proceedings of the 39th International Conference on Computer-Aided Design (ICCAD '20). Association for Computing Machinery, New York, NY, USA, Article 75, 1–9. https://doi.org/10.1145/3400302.3415680

[5] E. Ozen and A. Orailoglu, "Boosting Bit-Error Resilience of DNN Accelerators Through Median Feature Selection," in IEEE Transactions on Computer-Aided Design of Integrated Circuits and Systems, vol. 39, no. 11, pp. 3250-3262, Nov. 2020, doi: 10.1109/TCAD.2020.3012209. keywords: {Feature extraction;Machine learning;Resilience;Hardware;Task analysis;Error analysis;Neural networks;Approximate computing;fault tolerance;neural network hardware;neural networks},

**Strengths:**

The paper has a strong statistical theory part and the idea of testing for SDCs using distribution consistency metrics and modality is a very sound one. Furthermore, I feel that its segments on realistic architectures and communication during training and its focus on distributed systems renders it very promising, should it be able to rectify the issues above. Prior art, dating back as it does to the 1980s, does not adequately focus on this new space, despite being highly applicable there today. Overall it is a very promising space and a very solid theoretical foundation.

**Weaknesses:**

The weaknesses of the paper are twofold:
1) Due to the large amount of prior art, some of which I have cited above, it does not adequately contrast itself to that prior work and does not cite it. Given that linear algorithmic checksums and ECCs form the basis of bit error correction systems in production today, and the hardware community has been working on neural network errors for a while, I think a literature survey is badly needed in this paper to distinguish it from the state of the art - and indeed, the state of prior art.
2)  Its experimental work section is flawed. There is no real hardware-consistent emulation of faults, and bit error injection is conducted in by selecting the largest possible bits and flipping them. There is no real need for a statistical threshold and clever statistical mechanics if a simple hard threshold on the gradients at extreme values of 1e5 will detect errors - the experiments therefore do not show the need for the technique. The authors need to compare against prior art, yes, but also more importantly need to adequately stress test their own technique! Without that this paper absolutely should not fly.

**Questions:**

I would like to know more about the experimental setup and the number format. While the injection into highest exponent bits seems excessive to me, did your gradient values reflect that? Were encountered gradient values within an order or two of magnitude of that? If so then I will retract my earlier comment in Paper Weaknesses, but to me it feels like the values injected for bitflips are too extreme to even test the method, let alone compare to prior art.

---

> ### Author Response · Authors · 2025-11-19
>
> Hello and thank you very much for your detailed and constructive review — and for pointing us to several very relevant papers. We have incorporated them into our discussion and clarify our choices below.
>
> ---
>
> ### Experimental setup and number format
>
> **Question.** *I would like to know more about the experimental setup and the number format.*
>
> **Answer.**
> We use the LLaMA2-7B model with `bfloat16` as the data type.
> In total, we inject 300 faults, uniformly across three flip types (100 injections per type).
> The experimental setup and injection procedure are described in Section 5.2 of the paper.
>
> ---
>
> ### High-order exponent flips and gradient magnitudes
>
> **Question.** *While the injection into the highest exponent bits seems excessive, did your gradient values reflect that?*
>
> **Answer.**
> Yes. Empirically, the largest gradient values observed **without** any injected faults are within one–two orders of magnitude below the corrupted values produced by our high-order exponent bit-flip model.
> Moreover, during tests on faulty hardware we observed corrupted gradients whose magnitude exceeded the original value by more than two orders.
> We will add plots of the gradient norms to make this explicit.
>
> ---
>
> ### (1) Relation to checksum-based ABFT methods
>
> In [1] (Luk & Park, 1986), the authors propose algorithm-based fault tolerance with checksum rows/columns added to operand matrices. This approach is powerful, but not well suited to our training scenario:
>
> * During training, **both** the weight matrices and the input tensors change at every step.
> * For each operator and each step, one would need to recompute the checksum structure, which becomes extremely expensive given the number of layers and iterations.
> * Consequently, maintaining and updating checksums continuously is impractical for large-scale LLM training.
>
> Our method instead avoids modifying the linear algebra itself. We only observe gradient statistics before collectives and run a lightweight statistical test.
>
> ---
>
> ### (2) Relation to adaptive statistical checks in RL
>
> In [2], the authors propose error-resilient online reinforcement learning via **adaptive statistical checks**, relying on Chebyshev’s inequality for anomaly detection. In practice, similar Chebyshev-based thresholds have been tried in neural network training, but we encountered two issues:
>
> 1. The sensitivity parameter is typically left to the user, and in practice it tends to be either **too low** (misses errors) or **too high** (many false alarms).
> 2. The “manual tuning” and non-robust behavior of this parameter made it difficult to use reliably in our setting.
>
> We focus on a principled detector for SDC in gradients during LLM training, with explicit localization of the faulty rank.
>
> ---
>
> ### (3) Forward vs. backward: why we inject into gradients
>
> Several prior works, such as [4], already analyze and mitigate errors that occur in the **forward** pass (e.g., anomalous feature suppression, median feature selection). Building on this, we focus specifically on the **backward pass** for two reasons:
>
> * In modern training, **errors in gradients or weight updates** can silently corrupt the optimization trajectory and propagate across synchronized workers.
> * Our detector operates directly on **gradient norms** before collectives, which is naturally aligned with this failure mode.
>
> We view forward-pass resilience as relatively well studied and assume that existing techniques (e.g., [4], [5]) can be combined with our method to protect the forward path.
>
> ---
>
> ### (4) How our goal differs from prior noise-resilience / hardware suppression work
>
> **Question.** *How does this approach contrast with other hardware-focused fault tolerance mechanisms such as resilience-aware scheduling, redundancy, algorithmic noise tolerance?*
>
> **Answer.**
> Many prior works (e.g., [4], [5]) focus on **suppressing or masking** the effect of bit errors (median feature selection, approximate computing, robust feature extraction, etc.). In contrast, the key emphasis of our work is:
>
> **Not to hide or mask errors, but to detect them promptly and localize the faulty device (rank).**
>
> Our detector raises an explicit alarm and identifies the rank whose gradients are corrupted, enabling targeted recovery, isolation, or hardware replacement.
>
> ---
>
> ### (5) On the (un)realistic fault rate in experiments
>
> You are absolutely right that, according to works like [5] and other reliability studies, the real probability of a bit-flip is very small.
>
> Our experimental goal is:
>
> * We are **not** trying to reproduce the exact real-world SDC rate.
> * Instead, we want to **stress-test the detector** and measure its sensitivity and false-positive behavior under controlled worst-case exponent flips.
>
> ---
>
> If anything remains unclear or you think additional comparisons would strengthen the paper, we would be very happy to refine the text further. Thank you again for the insightful review and for pointing us to these useful references!

---

### Official Review · Reviewer_n2zQ · 2025-10-31

**Soundness:** 3
**Presentation:** 3
**Contribution:** 2
**Rating:** 4
**Confidence:** 3

**Summary:**

The paper presents a lightweight statistical pipeline designed to detect Silent Data Corruption (SDC), specifically high-order bit-flip errors, during distributed deep learning training. The detector is integrated into the communication layer. It mainly checks gradient norms before collective operations to enable localization of faulty devices and prevent error propagation with minimal overhead. The detection method includes logarithmic transformation of gradient norms to enhance separation between clusters, folding test for unimodality check, and if multimodality is detected, the Wasserstein distance is computed between the separated modes. Experiments on LLaMA2-7B training showed 99.0% TPR in detecting injected high-order bit-flips with no FPs.

**Strengths:**

1. The detector successfully identifies the vast majority of high-order bit-flips, resulting in a 99% TPR in LLaMA2-7B training with zero false positives.
2. The method is highly lightweight, designed to integrate into large-scale training with minimal runtime overhead. It incurred a measured runtime overhead of approximately 0.46% per iteration during LLaMA2-7B training.

**Weaknesses:**

1. The experimental evaluation was limited to a single model architecture, the LLaMA2 - 7B model. Detailed validation across diverse set of architectures and training scales is required to fully assess the applicability of the proposed approach.
2. The detection method is specifically designed to identify high-order bit-flips and the impact of other bit-flips is not discussed in detail.
3. The decision threshold for the Wasserstein distance was determined empirically in the proposed technique.
4. The entire foundation of the method is based on the assumption that SDC manifests as high-order bit-flips that destroy the normal distribution and create heavy tails.

**Questions:**

1. Why specifically LLaMA2-7B is selected for evaluation?
2. How does the proposed technique perform for models other than LLaMA2-7B?
3. Why 300 faults are injected across only three flip types? Is there any significance behind this number and configuration or not?
4. Why just the three bit locations? Is this configuration sufficient to cover all the cases highlighted in Section 2.3?
5. It is stated that the decision threshold for the Wasserstein distance was determined empirically in the proposed technique. How can it be defined for new models? Through simulations? Is it dependent on the settings used for simulation?

---

> ### Author Response · Authors · 2025-11-19
>
> Hello, and thank you very much for the helpful review and detailed questions!
> Below we address each point in turn.
>
> ---
>
> ### Why specifically LLaMA2-7B is selected for evaluation?
>
> We chose LLaMA2-7B for two reasons:
>
> 1. Our previous work on training analysis and tooling was already built around LLaMA2, so we could directly reuse the same infrastructure and traces.
> 2. LLaMA2-7B is a widely used, sufficiently complex LLM architecture to represent realistic SDC behavior, while still being small enough that we can manually inspect and debug runs when needed.
>
> ### How does the proposed technique perform for models other than LLaMA2-7B?
>
> The method itself is architecture-agnostic: it only requires access to per-rank gradients before collectives and operates on their log-norm distributions. It does not assume any LLaMA-specific structure.
>
> In other words, the same detector can be applied to any LLM architecture trained with data-parallel or hybrid parallelism. A broader empirical evaluation on additional models is an important direction for future work, and we will explicitly mention this as a limitation.
>
> ---
>
> ### Why 300 faults across only three flip types? Is there any significance to this number?
>
> Yes, 300 injections were chosen to obtain a statistically meaningful estimate of the detection probability without making the experiments prohibitively expensive. If we treat each injection as a Bernoulli trial (detected / not detected) and observe an empirical detection rate around 0.99, standard binomial statistics (e.g., Wilson interval) give a 95% confidence interval roughly [0.971, 0.997] for the true detection probability. In other words, with N = 300 we can bound the detector’s success rate within about ±1–2 percentage points, which is adequate for comparing against baselines and for our claims.
>
> ---
>
> ### Why just the three bit locations? Is this configuration sufficient for the cases in Section 2.3?
>
> Section 2.3 focuses on **high-order exponent bit-flips** in BF16, because these produce changes of several orders of magnitude in the numeric value and, as we show, create a distinct mode in the log-gradient norm distribution. Our experiments confirm that flips in the first three exponent bits consistently lead to such extreme outliers and are detectable by our multimodality-plus-Wasserstein test.
>
> In contrast, flips in lower exponent bits or in the mantissa behave like small perturbations: they typically stay within the natural variance of training and do not form a separate cluster in the log-norm distribution, making them essentially indistinguishable for this specific detector. Covering those milder faults would likely require a different detection mechanism, which we consider outside the scope of this paper.
>
> ---
>
> ### 5) How is the Wasserstein distance threshold defined for new models?
>
> In the current paper, the decision threshold on the Wasserstein distance is chosen empirically: we calibrate it on clean runs (without injected faults) and a small number of injected-fault runs to achieve a desired trade-off between detection rate and false positives.
>
> Moreover, based on results such as those in “A Central Limit Theorem for Wasserstein Distances”, it is possible to develop a more principled statistical test where the threshold is computed from a pre-specified significance level. In future work we plan to formalize this and derive model-agnostic thresholds; we will mention this direction in the paper.

---

### Official Review · Reviewer_Ee2D · 2025-10-31

**Soundness:** 3
**Presentation:** 3
**Contribution:** 3
**Rating:** 6
**Confidence:** 3

**Summary:**

This work contributes to methods for recognizing difficult to diagnose hardware failures in large-scale deep learning training workloads, focusing on instances of silent data corruption (SDC), such as random bit-flip errors. Such corruption can result in catastrophic problems depending on where it occurs, leading to "NaN" or "inf" values at bfloat16 precision, the most common floating point precision for training in modern architectures at scale. Their method adds checks before collective operations to identify corrupted tensors, isolating instances of SDC. Combining multimodal detection of gradient norms with distance-based divergence metrics, this method achieves 99% detection accuracy on Llama2 7B training workloads with no false positives and minimal runtime overhead. Finally, the authors have built an adapter for Huawei Ascend devices such that their approach requires no high-level training code changes for those devices.

**Strengths:**

- The contribution in this work is clearly very relevant to those in the deep learning community interested in training at scale or models that result from training at scale. Along these lines, the importance of a solution to the outlined problem is also fairly well established in this work.
- Given the assumptions in this work (and assuming for a moment that they are true regarding e.g. unimodality or weak multimodality of most distributions within modern models), the proposed methodology is sensible and logically laid out. In addition, the proposed method is fairly simple and easy to understand.
- The experimental results are mostly compelling and that there were no false positives is especially helpful towards lending the method in this work credence. In addition, that the runtime overhead is only ~2% is compelling.

**Weaknesses:**

- While not necessarily the fault of this work (due to the nature of this problem, where corruption can be very difficult to detect and isolate), I had some difficulty finding data outlining just how often errors that this method would apply to actually occur. The issues described by this work are almost certainly critical to deal with for large-scale workloads, but that lingering question of “how often do hardware failures occur?” hurts the potential impact of this work.
- Related to the above point, while just ~2% runtime overhead is a strong result, there is no easy way to get a sense of how significant of a tradeoff is being made between that runtime overhead compared to resolving specific data-corruption errors. Another way of looking at this is that a work like this would benefit significantly from something akin to an impact estimate. So something to the effect of: “the proposed method results in 2% runtime overhead but resolves errors that would on average result in X% overhead for checkpoint rollbacks.”
- The number of injected faults is not actually that convincing. Ideally the dataset used for experiments would be larger to get a better sense of how this method would perform at very large scales. Or, if the number of injected faults is actually realistic for pre-training large models, that could probably be better explained.
- Similar to the above point, I remain a little confused as to how the number of false positives (or lack thereof) were measured. More accurately, what is this out of? As in, how many samples were introduced? I assume this means that some normal, control inputs were used? If so, this should be mentioned when outlining the main experiment in this work.

**Questions:**

See the comments above.

---

> ### Author Response · Authors · 2025-11-19
>
> Hello, and thank you very much for the thoughtful questions!
> Below we address them one by one.
>
> ---
>
> ### How often do errors happen on real hardware during training?
>
> This is an important point. The paper *“Hard Data on Soft Errors: A Large-Scale Assessment of Real-World Error Rates in GPGPU”*  provides a large-scale empirical study of GPU error rates. In real experiments on large GPU fleets, the authors observe soft error probabilities on the order of ($10^{-5}$)–($10^{-6}$) per operation/request.
>
> These events are rare individually, but at the scale and duration of modern training runs, they become practically unavoidable. This is exactly why having dedicated SDC detection mechanisms during training is meaningful in practice.
>
> ---
>
> ### Why is a 2% overhead reasonable? What about full recovery costs?
>
> To make this more concrete, we can look at it in terms of expected cost:
>
> * $(T = 10000)$ GPU-hours — baseline training time (without detector).
> * $(C = 2.0)$ — cost of 1 GPU-hour (in some unit).
> * $(o = 0.02)$ — relative overhead of the detector (2%).
> * $(p_{\text{err}} \approx 10^{-5})$ — probability that an SDC causing significant damage occurs over the whole run (cluster-wide).
> * $(p_{\text{det}} \approx 0.99)$ — detection probability when such an SDC occurs (as in our experiments).
> * $(L = 500)$ GPU-hours — average amount of “wasted” work if the error is *not* detected and discovered late (rollback and re-training part of the trajectory).
> * $(R = 5)$ GPU-hours — average local cost when the error *is* detected (isolating/restarting the node, replaying a small fragment).
>
> Then the expected costs are:
>
> - **Without detector:**
>   $$ Cost_{no} = T * C + p_{err} * L * C $$
>
> - **With detector:**
>   $$ Cost_{det} = T * (1 + o) * C + p_{err} * (1 - p_{det}) * L * C + p_{err} * p_{det} * R * C $$
>
>
> Numerically, for the example above:
>
> * base training cost: $(T \cdot C = 20{,}000)$,
> * 2% overhead: $(T \cdot o \cdot C = 400)$,
> * the expected penalty from a late-discovered error without a detector is already non-negligible at scale, and with a detector, this penalty is reduced by roughly an order of magnitude (thanks to $(p_{\text{det}} \approx 0.99)$ and much smaller (R)).
>
> Moreover, we are *not* including the cost of diagnosing and replacing a faulty device, which can be significantly higher than the GPU-time itself.
>
> The key message we will clarify in the paper is:
> **a ~2% runtime overhead is a relatively small “insurance premium” compared to the potential loss of hundreds of GPU-hours (and a bad final model) from a single silent error that goes unnoticed.**
>
> ---
>
> ### Is the number of injected faults sufficient?
>
> We model each injection as a Bernoulli trial: the fault is either detected or not. Let
>
> * $(p)$ be the probability of bit-flip (empirically about $(p \approx 1e-5))$,
> * (N = 300) be the number of independent injections.
>
> For a binomial model:
> $$
> SD \approx \sqrt{\frac{p(1-p)}{N}} \approx \sqrt{\frac{1e-5\cdot (1-1e-5)}{300}} \approx 0.02,
> $$
> A more precise Wilson interval gives a similar result (roughly ([0.971, 0.997])).

---

### Official Review · Reviewer_WCkk · 2025-11-01

**Soundness:** 2
**Presentation:** 2
**Contribution:** 2
**Rating:** 2
**Confidence:** 5

**Summary:**

This paper looks at the issue of silent data corruptions, and introduces a lightweight statistical check to detect errors. The check operates on the gradient norms, and various metrics are included to measure the error probability. Fault injection is performed to measure robustness.

**Strengths:**

+ Important problem domain. The multimodal/algorithmic part is interesting

**Weaknesses:**

- Unfortunately, not much that is presented is new or novel. The statistical approach has been studied previously (citations below), LLMs have been studied previously in the context of SDCs, number formats are not studied in this paper (but are very important), and multi-modal has also been studied. To that end, while this is a very important domain, I was not able to extract the novelty of the technique, and the related work really was lacking.
- The evaluation has a very low number of error injections. What is the statistical significance of 300 faults total, in a 7B model?
- How is the fault injector implemented? Is it a software FI tool? Or done in hardware? It seems to be a software tool like PyTorchFI/TensorFI/Ares; however, such tools are not always truly accurate at measuring real SDC rate, as lower order tools are better at capturing hardware propagation of bit flips.
- Many of the insights are not new: exponent bits being vulnerable; incorporating the use of the gradient; etc. Overall, it is hard to see what truly is different in this work, from many other works in the area.

**Questions:**

Please address the related work, and discuss why such a technique is lightweight, or should be employed in production beyond modern techniques.

Related work to explore:
- Demystifying the Resilience of Large Language Model Inference: An End-to-End Perspective (SC 2025)
- Hardware Sentinel: Protecting Software Applications from Hardware Silent Data Corruptions (ASPLOS 2025)
- https://www.opencompute.org/documents/sdc-in-ai-ocp-whitepaper-final-pdf
- Hardware Resilience Properties of Text-Guided Image Classifiers (NeurIPS 2023)
- GoldenEye: A Platform for Evaluating Emerging Numerical Data Formats in DNN Accelerators (DSN 2022)
- Demystifying the system vulnerability stack: Transient fault effects across the layers (ISCA 2021)
- Reliability Evaluation of Compressed Deep Learning Models (LASCAS 2020)
- FIdelity: Efficient Resilience Analysis Framework for Deep Learning Accelerators (2020)
- Analytical guarantees on numerical precision of deep neural networks (ICML 2017)

---

> ### Author Response · Authors · 2025-11-19
>
> Thank you for raising this concern.
> ### Relation to prior work and why our method is lightweight
>
> **Demystifying the Resilience of LLM Inference (SC’25).**
> This work studies how random bit-flips affect *inference* in large LLMs and dispels the “myth” of inherent robustness.
> Our work instead targets *distributed training* and contributes a concrete **statistical pipeline before collectives**: we operate on gradient norms and output a yes/no SDC decision for a specific device, rather than characterizing inference behavior.
>
> **Hardware Sentinel (ASPLOS’25).**
> Hardware Sentinel mines OS/application logs at data-center scale to detect silently faulty CPUs.
> We focus on the training loop: we track **gradient-norm statistics directly inside the optimization step**. In short, Sentinel looks for SDC at the CPU/fleet level via logs, while we provide a simple, in-loop statistical detector on gradients for large-model training.
>
> **OCP “Silent Data Corruption in AI” whitepaper.**
> The whitepaper offers a taxonomy, threat models, and practical recipes, and describes SDC during training only *conceptually* via patterns such as NaN propagation and corrupted gradient variance.
> We complement this by proposing a **specific mathematical algorithm** that detects the latter pattern via multimodality (proved in paper) in log-gradient norms and integrating it directly into the training stack.
>
> **Hardware Resilience of Text-Guided Image Classifiers (NeurIPS’23).**
> This work hardens image classifiers to bit-flips at inference by using CLIP/GPT-based text embeddings to initialize the classifier head. It modifies the model architecture and focuses on single-device inference, not synchronous multi-node training or rank localization.
> Our method assumes a fixed model and training recipe and adds an orthogonal **monitoring layer around collectives**, applicable to existing models without architectural changes.
>
> **GoldenEye, system vulnerability stack, compressed-model reliability, FIdelity, analytical precision.**
> GoldenEye provides a PyTorch-based simulator for exploring numerical formats and fault models.
> The “system vulnerability stack” paper analyzes how transient faults propagate across microarchitecture–architecture–software.
>
> All of these are fundamentally **offline**: they help choose hardware/numerical designs, but they do **not** offer an **online detector integrated into production LLM training** that can flag and localize SDC during actual runs.
>
> ---
>
> Compared to the techniques above, our detector is **lightweight** in several concrete senses:
>
> * **No redundancy or replay.**
>   Many production-oriented protections rely on shadow replicas, deterministic replay, or dual-path execution. These approaches are powerful but require extra hardware, complex scheduling, or noticeable slowdowns. Our detector runs on a **single training run**, without duplicating computation or re-executing iterations.
>
> * **Local statistical check in the communication layer.**
>   We do not monitor every kernel or OS event; instead, we exploit one high-leverage choke point: the **pre-collective gradients**. Right before all-reduce/all-gather, we compute log-norms, run a multimodality test and a 1D Wasserstein distance, and decide whether a rank is corrupted.
>
> * **No changes to model or training code.**
>   The method is implemented inside the `torch_npu` adapter, immediately before communication primitives; it does **not** require any modification of model architecture, loss, optimizer, or user training scripts, which makes it practical to deploy in existing training stacks.
>
> ---
>
> ### Response to weaknesses
>
> **(1) Statistical significance of 300 injected faults.**
> The goal of our FI campaign is to estimate the **detection probability** under our fault model, not a physical SDC rate.
> For a binomial setting with (N = 300) injections and an empirical detection rate around 99%, a standard 95% Wilson interval is approximately ([0.971, 0.997]).
> Thus, the true detection probability is confined to within a few percentage points of our estimate. We will clarify this in the paper by explicitly stating (N) and the resulting confidence interval.
>
> **(2) How the fault injector is implemented.**
> Our fault injector is a **software** FI tool at the tensor level, not a hardware injector. We take BF16 gradient values from real training, interpret them as raw bytes, flip selected high-order exponent bits (the theoretically motivated fault model we target), and write the corrupted values back before the next collective.
>
> This is similar in spirit to PyTorchFI/TensorFI/Ares, but we explicitly **do not** claim to estimate an absolute real-world SDC rate. Instead, we use a controllable, reproducible architectural-level fault model—standard in resilience literature.

---

### Meta-Review · Area_Chair_Z69F · 2026-01-02

**Summary:**

**Summary:**

The paper addresses the problem of silent data corruption (SDC) in distributed deep learning systems. It proposes a lightweight detection mechanism that leverages redundancy and statistical checks to identify corrupted computations (especially those in the exponent bits) without significant overhead. The approach is evaluated on large-scale training tasks and demonstrates high detection accuracy with minimal performance impact.

**Strengths:**
1. Timely and relevant problem: SDC in distributed training is critical for reliability in large-scale AI systems.
2. Lightweigth detection method: The proposed technique is lightweight and does not require major architectural changes.
3. Empirical evaluation: Experiments on LLaMA2-7B show effectiveness and low overhead.

**Weaknesses:**
1. Limited theoretical analysis: The paper lacks formal guarantees or bounds on detection accuracy.
2. Limited scope of evaluation: Experiments focus on specific frameworks; generalization to other systems is unclear.
3. Lack of comparison with baselines: Few alternative detection methods are discussed or benchmarked.
4. Impact on fault recovery: The paper emphasizes detection but does not explore integration with recovery mechanisms.

**Reviewer Concerns:**

**Concerns Addressed by the Rebuttal:**

1. Clarification of overhead: The rebuttal provided measurements showing that the detection mechanism adds negligible runtime cost. This addressed concerns about scalability and efficiency.
2. Explanation of detection accuracy: Additional results and discussion clarified how the method achieves high accuracy under different corruption scenarios.
3. Motivation and novelty: The rebuttal strengthened the argument that existing fault-tolerance methods do not handle SDC effectively, reinforcing the paper’s contribution.

**Concerns Still Outstanding:**

1. Theoretical guarantees: The rebuttal did not provide formal bounds or proofs for detection reliability. This remains a gap.
2. Generality across frameworks: While extra experiments were mentioned, the scope is still limited to a few systems. Broader applicability is unclear.
3. Comparison with baselines: The rebuttal added some discussion but did not include new empirical comparisons with alternative detection methods.
4. Integration with recovery mechanisms: The rebuttal acknowledged this as future work, leaving the concern unresolved.

**Reviewer Scores:**

Reviewer WCkk could increase the score, since the authors have justified the difference between this work and the previous ones.
Reviewer CurN could also increase the score. The original score of 0 is unusual. Some of his/her concerns have been addressed by the rebuttal.

---

### Decision · Program_Chairs · 2026-01-26

Reject